# Novel Purine Chemotypes with Activity against *Plasmodium falciparum* and *Trypanosoma cruzi*

**DOI:** 10.3390/ph14070638

**Published:** 2021-07-01

**Authors:** Nieves Martinez-Peinado, Álvaro Lorente-Macías, Alejandro García-Salguero, Nuria Cortes-Serra, Ángel Fenollar-Collado, Albert Ros-Lucas, Joaquim Gascon, Maria-Jesus Pinazo, Ignacio J. Molina, Asier Unciti-Broceta, Juan J. Díaz-Mochón, María J. Pineda de las Infantas y Villatoro, Luis Izquierdo, Julio Alonso-Padilla

**Affiliations:** 1Barcelona Institute for Global Health (ISGlobal), Hospital Clínic—University of Barcelona, 08036 Barcelona, Spain; nieves.martinez@isglobal.org (N.M.-P.); alejandro.garsal@hotmail.com (A.G.-S.); nuria.cortes@isglobal.org (N.C.-S.); angel.fenollar@isglobal.org (Á.F.-C.); albert.ros@isglobal.org (A.R.-L.); quim.gascon@isglobal.org (J.G.); mariajesus.pinazo@isglobal.org (M.-J.P.); 2Department of Medicinal & Organic Chemistry and Excellence Research Unit of “Chemistry Applied to Biomedicine and the Environment”, Faculty of Pharmacy, University of Granada, Campus de Cartuja s/n, 18071 Granada, Spain; alvaro.lorente@ed.ac.uk (Á.L.-M.); juandiaz@ugr.es (J.J.D.-M.); 3Institute of Biopathology and Regenerative Medicine, Centre for Biomedical Research, University of Granada, Avda. del Conocimiento s/n, 18100 Granada, Spain; imolina@ugr.es; 4Cancer Research UK Edinburgh Centre, Institute of Genetics & Cancer, University of Edinburgh, Crewe Road South, Edinburgh EH4 2XR, UK; asier.ub@ed.ac.uk

**Keywords:** *Plasmodium falciparum*, *Trypanosoma cruzi*, purine metabolism, purine derivatives, pyrimidine analogs, phenotypic assays, cytotoxicity assays

## Abstract

Malaria and Chagas disease, caused by *Plasmodium* spp. and *Trypanosoma cruzi* parasites, remain important global health problems. Available treatments for those diseases present several limitations, such as lack of efficacy, toxic side effects, and drug resistance. Thus, new drugs are urgently needed. The discovery of new drugs may be benefited by considering the significant biological differences between hosts and parasites. One of the most striking differences is found in the purine metabolism, because most of the parasites are incapable of de novo purine biosynthesis. Herein, we have analyzed the in vitro anti-*P. falciparum* and anti-*T. cruzi* activity of a collection of 81 purine derivatives and pyrimidine analogs. We firstly used a primary screening at three fixed concentrations (100, 10, and 1 µM) and progressed those compounds that kept the growth of the parasites < 30% at 100 µM to dose–response assays. Then, we performed two different cytotoxicity assays on Vero cells and human HepG2 cells. Finally, compounds specifically active against *T. cruzi* were tested against intracellular amastigote forms. Purines **33** (IC_50_ = 19.19 µM) and **76** (IC_50_ = 18.27 µM) were the most potent against *P. falciparum*. On the other hand, **6D** (IC_50_ = 3.78 µM) and **34** (IC_50_ = 4.24 µM) were identified as hit purines against *T. cruzi* amastigotes. Moreover, an in silico docking study revealed that *P. falciparum* and *T. cruzi* hypoxanthine guanine phosphoribosyltransferase enzymes could be the potential targets of those compounds. Our study identified two novel, purine-based chemotypes that could be further optimized to generate potent and diversified anti-parasitic drugs against both parasites.

## 1. Introduction

Infectious diseases caused by protozoan parasites, such as malaria and Chagas disease, remain an important global health problem, causing high morbidity and mortality in humans. Malaria, caused by *Plasmodium* spp. parasites, affects more than 200 million people worldwide, and inflicts the highest burden in Africa, mostly in children under five years old [1]. Amongst the five *Plasmodium* species that cause malaria in humans, *P. falciparum* is the deadliest, and the most prevalent in Africa [1,2]. Artemisinin-based combination therapies (ACTs) are the recommended first-line treatments against malaria [3]. However, parasite resistance against artemisinin has considerably increased, and become widespread in the last few years [4,5,6].

Chagas disease, whose etiological agent is the protozoan parasite *Trypanosoma cruzi*, affects more than 6 million people worldwide [7]. The disease is endemic in Latin America, but has spread its impact to non-endemic countries due to migrations in recent decades [8]. Chemotherapeutic treatment is limited to two drugs—benznidazole (BNZ), and nifurtimox (NFX)—which present variable efficacy at the chronic stage of the disease, along with frequent side effects [9,10]. Since available drugs to treat these infectious diseases have several shortcomings, the discovery of new active compounds against *P. falciparum* and *T. cruzi* is urgently needed [11].

The increasing knowledge of the parasites’ biology, together with the availability of their complete genomes, has made the targeted design of new compounds a promising drug discovery strategy [12]. Exploiting biochemical and physiological differences between parasites and hosts could contribute to the development of new drugs. One of the most outstanding divergences between these can be found in the purine metabolism [12]. As with other protozoan parasites, *P. falciparum* and *T. cruzi* are incapable of de novo biosynthesis of purines. Consequently, they depend on the purine salvage pathway for development and proliferation [13,14]. The purine salvage pathway entails transmembrane transporters for the uptake of nucleosides/nucleobases, as well as a series of processing enzymes, which show considerable differences between parasites and mammals—some of them not being present in the latter. Such enzymes thus constitute very attractive chemotherapeutic targets [15]. In this context, two different approaches have been evaluated: the inhibition of specific enzymes, and the use of subversive substrates.

*P. falciparum* purine metabolism mainly relies on three enzymes: hypoxanthine-guanine-xanthine phosphoribosyltransferase (HGXPRT), adenosine deaminase (ADA), and purine nucleoside phosphorylase (PNP) [16]. Within the parasite, purine salvage metabolism is funneled through hypoxanthine by the sequential actions of ADA and PNP, prior to phosphoribosylation by HGXPRT [16]. ADA and PNP have dual catalytic specificities that allow them to use purine and methylthiopurine substrates [16]. Most reports have focused on targeting the purine nucleoside phosphorylase (PNP) of *P. falciparum* as a mechanism to disrupt two metabolic pathways by targeting just one enzyme [16], or on the inhibition of its HGXPRT to avoid the conversion of hypoxanthine, guanine, or xanthine into their monophosphate derivatives [17]. Those enzymes are key players in *P. falciparum*’s purine metabolism, but there are others—such as guanosine-5′-monophosphate synthase (GMPS), adenylosuccinate synthetase (ADSS), and inosine-5′-monophosphate dehydrogenase (IMPDH)—that are also involved [12].

The purine salvage pathway in *T. cruzi* includes the following relevant enzymes from a chemotherapeutical point of view: adenine phosphoribosyl transferase (APRT), hypoxanthine-guanine phosphoribosyltransferase (HGPRT), adenosine kinase (AK), methylthioadenosine phosphorylase (MTAP), and nucleoside hydrolases (NHs) [12]. *T. cruzi* parasites salvage hypoxanthine and guanine in preference to adenosine, which are converted to IMP and GMP by the enzymatic activity of HGPRT [14]. In addition, hypoxanthine is the most abundant purine base in human serum. Thus, HGPRT plays a crucial role in the survival of *T. cruzi* parasites, which has made it the target chosen in most of the studies performed [18,19,20,21].

Others have screened collections of purine- or pyrimidine-based nucleosides without focusing on a specifically targeted enzyme [22,23,24]. This is the strategy that has been followed in the present work. Recently, some β-hydroxy- and β-aminophosphonate acyclonucleosides have been shown to inhibit *P. falciparum* growth [25], while 7-deazaguanosine compounds displayed activity against *Leishmania donovani*, *Trypanosoma brucei*, *T. cruzi* and *P. falciparum* [26]. Similarly, purine derivatives such as tubercidin, formycin A, formycin B, cordycepin, puromycinaminonucleoside, and allopurinol have shown to have in vitro anti-*T. cruzi* activity [24]. Moreover, allopurinol, whose active metabolite is generated by phosphoribosylation, has even reached clinical evaluation for Chagas disease [27].

Herein, we have phenotypically evaluated the anti-*P. falciparum* and anti-*T. cruzi* activity of a collection of 81 purine derivatives and pyrimidine analogs that were developed via different synthetic procedures focused on the exploration of positions 6, 8, and 9 of the purine ring (Figure 1) [28,29,30]. With the prioritized compounds specifically active against each parasite, we additionally performed a computational docking study to identify their potential targets.

## 2. Results

### 2.1. Primary Screening

Upon screening the collection of 81 compounds at three concentrations (100, 10, and 1 µM), *P. falciparum* growth was kept below 30% relative to the control in the presence of 10 of the compounds at 100 µM, and also of 1 of them when evaluated at 10 µM (Table 1). Among those compounds, 2 belonged to group C and 8 to group D (Appendix A). On the other hand, *T. cruzi* growth was kept below 30% of its control when exposed to 28 of the compounds at 100 µM, and also when exposed to 2 of them at 10 µM (Table 1). In contrast to what was observed in *P. falciparum*, at least one compound per group showed activity against *T. cruzi*. Indeed, 2 compounds were from group A, 2 from group B, 5 from group C, 17 from group D and 2 from group E (Appendix A). Neither parasite exhibited growth inhibition above 50% when compounds were at 1 µM (Table 1). Figure 2 depicts some representative examples of the percentage of parasite growth in the presence of compounds from different structural groups.

### 2.2. Dose–Response Growth Inhibition Assay

A total of 10 and 28 compounds were selected to progress to dose–response assays against *P. falciparum* and *T. cruzi*, respectively, based on the criteria of keeping the growth of the parasites below 30% at 100 µM. The compounds’ potency (IC_50_) was determined following a 1:2 dilution pattern to conform to dose–response curves. In every round of the anti-*T. cruzi* assays, the reference drug BNZ was included as a control of drug growth inhibition; it showed an average IC_50_ value of 1.72 (0.13) µM, which correlates with that of previous reports [31,32]. In the case of *P. falciparum*, fixed concentrations of chloroquine were used as a control of drug growth inhibition in different assays.

The IC_50_ values of the 10 compounds tested against *P. falciparum* ranged from 18.3 to 110.1 µM (Table 2). Compounds **6g**, **33**, **60**, and **76** showed the highest activity, with IC_50_ values around 20 µM (Table 2; Figure 3). It is of interest to note that those compounds were from structurally diverse subgroups (groups C, D1, D5, and D6). Moreover, a wide range of IC_50_ values was observed between subgroups (Table 2).

Similarly, IC_50_ values derived from the anti-*T. cruzi* assays ranged from 0.94 to 109.90 µM (Appendix A). We determined an IC_50_ cutoff threshold of 10-fold that of the BNZ IC_50_ value, in order to progress only those compounds with high activity against the parasite. Consequently, eight compounds were identified; among those, there were none from groups A or B, while compounds **94**, **93**, and **6g**—from subgroups D7, E3, and C2, respectively—exhibited the highest activity, with IC_50_ values of 0.92 (0.15), 1.42 (0.19) and 3.97 (0.32) µM, respectively (Table 2; Figure 4).

### 2.3. Identification of Compounds with Specific Antiparasitic Activity

In order to discard those compounds that did not show specific activity against *T. cruzi,* we performed a secondary Vero cell toxicity assay [31]; we also included in this assay those compounds that were active against *P. falciparum*, in order to assess their toxicity against a mammalian cell line.

We included the reference drug BNZ, which reached an average TC_50_ value of 209.5 (15.0) µM, similarly to what had been reported before [31]. At the concentration range evaluated, which was determined by the maximum percentage of DMSO permitted by the assay, all compounds had low toxicity to Vero cells. This resulted in flattened curves that did not allow us to accurately calculate the TC_50_ values. For this reason, we report the minimum value of the 95% CI, according to which compounds **25** and **34** showed the lowest toxicity to Vero cells, followed by compounds **93**, **29** and **32** (Table 2). Interestingly, three of them belonged to subgroup D1.

Regarding the specificity of the activity of the compounds against *P. falciparum* and *T. cruzi*, most of them yielded high selectivity index (SI) windows, with values depicting their cytotoxicity contribution as being insignificant. Generally, SI values > 10 are suitable for progression in the screening cascade [33]. Five of the ten compounds active against *P. falciparum* displayed SI values > 10 (Table 2). Among those, compound **25** reached the highest SI index, with a value > 73.6 (Table 2). Compounds **6D**, **6g**, **59**, **84**, and **88**—with SI values < 10—were discarded from further progression. On the other hand, seven of the eight compounds active against *T. cruzi* had SI values > 10. Indeed, compounds **29**, **34**, and **93** showed higher SI values than that of the standard drug BNZ itself (Table 2). Only compound **70** had an SI < 10 against *T. cruzi* (Table 2).

### 2.4. HepG2 Cell Toxicity Assay

We also assessed the toxicity of the active compounds against both parasites in an assay based on human HepG2 cells, which are a commonly used cell-based model to anticipate potential liver toxicity [34,35]. The TC_50_ of the reference drug BNZ was 255.3 (10.8) µM, which was similar to that previously reported [31]. As mentioned above, we reported the HepG2 toxicity as the minimum value of the 95% CI since they all had a low toxicity. All compounds tested against HepG2 cells showed lower toxicity to this cell line than to Vero cells (Figure 5; Table 2), and only compound **6D** turned out to be more toxic than BNZ to HepG2 cells (Table 2). Representative dose–response curves of some of the compounds tested against Vero and HepG2 cells are shown in Figure 5.

### 2.5. Antiamastigote Growth Inhibition Assay

We next tested those compounds that showed specific activity against *T. cruzi* with an assay carried out on intracellular amastigotes, which are considered to be the main targets for any prospective drug against chronic *T. cruzi* infections. The reference drug BNZ was included in each antiamastigote assay, and reported an average IC_50_ value of 1.82 (0.08) µM. Overall, the compounds’ specific antiamastigote activity was lower than that observed in the antiparasitic assay previously described, excluding that of compound **6D**.

In fact, only compounds **6D** and **34** complied with the IC_50_ cutoff threshold of 10-fold that of BNZ, reaching IC_50_ values of 3.78 (0.31) µM and 4.24 (0.29) µM, respectively (Table 2, Figure 6). Both compounds yielded SI values > 10 against Vero cells, with that of compound **34** being outstanding (SI > 1415.10; Table 2).

### 2.6. Computational Analysis

Biological assays led to the identification of two hit compounds (2.5% hit rate) against each parasite. In order to identify potential targets of those compounds, we performed an in silico docking study with enzymes from the purine salvage pathways of both parasites. The lowest free binding energy (i.e., best docking score) indicates the highest ligand/protein affinity. The docking study with the compounds was conducted in comparison with enzymes’ natural ligands, as these compete for the active binding site. In general, compounds showed lower binding energy values than natural ligands (Table 3 and Table 4). Compounds **33** and **76** reported the lowest binding energy values when docked against the *Pf*HGXPRT enzyme (Table 3). Indeed, the energy difference between both compounds and natural ligands was the highest among docked enzymes (Table 3). Both compounds also reported considerably higher energy differences when docked against *P. falciparum* adenosine deaminase (*Pf*ADA). Even though *P. falciparum* adenylosuccinate synthetase (*Pf*ADSS) reported low binding energy values for compounds, it also did so for its natural ligand—inositol monophosphate (IMP; ∆G = −9.49 (0.03) Kcal/mol). Regarding hit compounds against *T. cruzi*, **6D** reported the lowest binding energy values with *T. cruzi* adenosine kinase (*Tc*Ak; ∆G = −9.20 (0.00) Kcal/mol) and *T. cruzi* inosine-guanosine nucleoside hydrolase (*Tc*IGNH; ∆G = −9.10 (0.02) Kcal/mol), while **76** did so with *T. cruzi* inosine-adenosine-guanosine nucleoside hydrolase (*Tc*IAGNH; ∆G = −10.09 (0.17) Kcal/mol) and *T. cruzi* adenylosuccinate lyase (*Tc*ADSL; ∆G = −10.08 (0.32) Kcal/mol). However, contrary to *P. falciparum* docking results, the difference in binding energy values between the compounds and the natural ligands of those enzymes was not the highest (Table 4). The highest energy difference value of both compounds was shown for *Tc*HGPRT (Table 4).

For both parasites, *Pf*HGXPRT and *Tc*HGPRT docking results were chosen for visualization due to their high energy differences between the binding of compounds and natural ligands. Such visualization revealed that the compounds bind to the active site occupying more space of the cavity than the natural ligands of both enzymes (Figure 7A,B and Figure 8A,B).

In both cases, hypoxanthine is located in a hydrophobic area of the binding pocket. Interestingly, an aromatic group of each compound is located in the binding site of hypoxanthine, while the rest of the molecule extends to the pyrophosphate, ribose, and 5´-phosphate binding sites (Figure 7 and Figure 8). In all cases, electronegative atoms of the purine base tend to locate in a more electronegative and hydrophilic area of the binding site (Figure 7A,B and Figure 8A,B).

In *Pf*HGXPRT, compound **33** is found entirely inside the cavity, while in compound **76** the substitution at C6 remains out (Figure 7A,B). Hypoxanthine forms four hydrophobic interactions and five H bonds with aspartic (D) 148, lysine (K) 176, and valine (V) 198. Residues of tyrosine (T) 116 and phenylalanine (F) 197 are involved in a pi–pi stacking interaction (Figure 7C). Compound **33** was found to have eight hydrophobic interactions, sharing three of them with hypoxanthine. In compound **33**, two nitrogen atoms from the triazole group form two hydrogen bonds with Y116, and the oxygen at position C6 interacts by forming two hydrogen bonds with threonine (T) 152 and serine (S) 115 (Figure 7C). Hydrophobic interactions in compound **76** were observed with isoleucine (I) 146, F197, asparagine (N) 206, and leucine (L) 207. In compound **76**, the nitrogen from the 1-thiazol-4-ylmethyl group and the N1 from the purine ring interact with K114 via two hydrogen bonds (Figure 7C). Interestingly, there is no hydrogen bond shared between hypoxanthine and the compounds, whereas hydrophobic interactions that involve I146 and F197 are present in all three bindings, and the interaction with Y116 in the binding of hypoxanthine and compound **33**.

In *Tc*HGPRT, the 6-benzoxy and 8-phenyl substitutions of compounds **6D** and **34** occupied the location of hypoxanthine. Hypoxanthine forms two hydrophobic interactions with L170 and I113, and three hydrogen bonds with D115, K143, and V165 (Figure 8C). Compound **6D** only forms one hydrogen bond with arginine (R) 177, but was found to have seven hydrophobic interactions—those with I113, L170, and F164 being shared with hypoxanthine (Figure 8C). On the other hand, compound **34** forms hydrophobic interactions with V92, I113, and F164, and hydrogen bonds with residues T119, L118, and T116 (Figure 8C). Similarly to *Pf*HGXPRT bindings, there is no hydrogen bond shared between hypoxanthine and the compounds, while the hydrophobic interaction with isoleucine I113 and the pi–pi stacking interaction with F164 are present in all three bindings.

## 3. Discussion

Targeted design of new drugs is an interesting strategy to specifically inhibit key enzymes diverging between the parasite pathogens and their hosts. Most of the protozoan parasites, including *P. falciparum* and *T. cruzi*, depend on the purine salvage pathway. Thus, targeting the purine metabolism could be a valuable approach for the design of new drugs against these infectious diseases. In this study, we screened a synthetic collection of 81 structurally related purines and pyrimidines in a target-agnostic manner, in order to identify those with the highest specific antiparasitic activity in vitro. Notably, from a structural standpoint, compounds within this collection can be classified in five distinct chemical groups [36].

Among those compounds that were specifically active against both parasites, we found three belonging to chemical group C, characterized by the presence of a benzoxy group at position C6 of the purine ring (Figure 1). Compound **29**, from subgroup C1, was the fifth most potent compound (IC_50_ = 4.86 (0.65) µM) against *T. cruzi**;* this compound features a methylpiperazine group at N9 and no substitution at C8 (Figure 1). Compounds **6D** and **6g**, from subgroup C2, showed activity against both parasites, but reported an SI value < 10 against *P. falciparum*. Compound **6g** was the third most potent compound (IC_50_ = 3.97 (0.32) µM) against *T. cruzi*, showing a specific antiparasitic activity in comparison to Vero and HepG2 cells; in this compound, the three positions of the purine ring are substituted with an aromatic group (Figure 1). Compound **6D**, with a tert-butyl instead of an aromatic substitute group at N9, showed lower activity against *T. cruzi* (IC_50_ = 5.56 (0.33) µM, Table 2). Nevertheless, it was the only compound from this group with specific antiamastigote activity (IC_50_ = 3.78 (0.31) µM; Table 2). 6-Benzoxy purines with voluminous groups at N9 were more active against *T. cruzi* than those with small substituents in the anti-*T. cruzi* assay, but did not show activity against amastigote forms. In addition, the presence of a phenyl group at C8 seems to play an important role. In this regard, Hulpia et al. reported promising in vitro and in vivo activities of certain C7 phenyl-substituted analogs against *T. cruzi* [24]. The use of aromatic groups, such as the ones presented in those compounds, is recurrent in the design of purine inhibitors [23,37]. For instance, Harmse et al. demonstrated that the presence of a voluminous aromatic ring in the purine structure had antimalarial activity [37]. In addition, Singh et al. reported IC_50_ values < 5 µM for nucleoside homologs stabilized with aromatic rings against *P. falciparum* [23]. Moreover, our own works demonstrate that a compound synthesized by similar synthetic strategies and featuring a 6-benzoxy group at C6 position has high activity against *T. brucei* (IC_50_ = 1 (0.1) µM) [28].

Compounds belonging to group D are characterized by the presence of an *N*-substituted triazolylmethoxy group at position C6 of the purine ring (Figure 1). All compounds that were specifically active against *P. falciparum* belonged to this group, highlighting the relevance of this chemical substituent against *P. falciparum*. The closely related structure of these compounds suggests a common target in *P. falciparum*. Moreover, the activity of triazole-based compounds has already been reported against this parasite [38]. For instance, triazolopyrimidine hybrids were identified as being potent in vitro and in vivo inhibitors of *P. falciparum* growth [38]. The role of the triazole group against *T. cruzi* is not so clear, since just four out of seven compounds present this substituent. However, triazole-based drugs have also been described as possessing high in vitro activity against *T. cruzi* [39,40].

Taking a deeper insight into chemical group D, we found that active compounds against both parasites belonged to subgroups D1, D5, D6, and D7. Active compounds against *P. falciparum* from subgroup D1 principally differ in the position of the methoxy group attached to the phenyl ring that substitutes the triazole (Figure 1). The anti-*P. falciparum* potency of compounds **25**, **32**, and **33** ranged from 19.2 to 81.5 µM, and increased in relation to bearing a methoxy group in the positions *para* < *meta* < *ortho*. Interestingly, the toxicity of those compounds to Vero cells also increased following the same chemical substitution pattern. From this subgroup, only compound **34** showed activity against *T. cruzi* (IC_50_ = 4.12 (0.41) µM), yielding the highest SI rate against Vero cells (SI > 1456.31) (Table 2); this activity was kept against amastigote forms (IC_50_ = 4.24 (0.29) µM, SI > 1415.10) (Table 2). Compound **34** shares a phenyl group at C8 with compound **6D**, which is the only other compound displaying specific antiamastigote activity; moreover, both compounds have a small substituent at N9.

In subgroup D5, only compound **60** showed specific activity against *P. falciparum*, while compound **59** was discarded due to its low selectivity with respect to Vero cells. Although the toxicity of both compounds to Vero cells was similar, the IC_50_ value of compound **60** was around five times lower than that of compound **59** (Table 2); structurally speaking, the main difference between these compounds is that compound **60** carries a 2-methoxyphenyl group at position N1 of the triazole, while **59** displays a 3-methoxyphenyl group (Figure 1). Thus, as occurs with chemical subgroup D1, a similar trend between structure and activity is observed. Interestingly, compounds belonging to subgroup D5 are the pyrimidine Schiff base analogs of those from subgroup D1. There were no statistically significant differences in the effects on *P. falciparum* growth between the active compounds and the correspondent pyrimidine analogs. For example, compound **33** had an IC_50_ = 19.19 (1.1) µM, while its pyrimidine Schiff base analog **60** showed an IC_50_ = 23.3 (1.06) µM (Table 2). It is of interest to note that almost all compounds from subgroup D5 reported activity against *T. cruzi*, even though their IC_50_ values were > 10-fold higher than the BNZ IC_50_ value and, thus, were not progressed to toxicity assays.

Compound **76**, from subgroup D6, was the most potent of the collection against *P. falciparum* (IC_50_ = 18.27 (1.07) µM; Table 2). Compound **84**, with an IC_50_ value 1.5 times higher against this parasite, had an SI < 10 against Vero cells, and was thus discarded. While both compounds possess a thiazole-4-yl-methyl group connected at the triazole moiety at C6 of the purine ring, and a phenyl group at C8, they differ in position N9, where **76** features a piperidine ring and **84** a methylpiperazine (Figure 1). That increased basic character of **84** may be involved in the toxicity increase against Vero cells. The high activity of compound **76** may be related to the voluminous substituent at position N9, which could fit in the ribose pocket of the target enzyme(s). From this subgroup, only compound **70** displayed activity against *T. cruzi*, although it was toxic to Vero cells, and discarded for its SI < 10 (Table 2).

Within subgroup D7, we found that compound **88** presented activity against both parasites; it has a similar structure to that of compound **32**, but no substitution at C8, and a tert-butyl at position N9, yet it showed an IC_50_ value close to that observed for **32** against *P. falciparum*. Nevertheless, it had an SI < 10 against this parasite, and was thus discarded. On the other hand, this same compound showed an IC_50_ value of 12.72 µM against *T. cruzi* (Table 2). Compound **94** was the most potent of the collection in the anti-*T. cruzi* assay (IC_50_ = 0.92 (0.15) µM; Table 2), and also belonged to the same chemical group; it shared with **88** the 1-(3-methoxyphenyl)-1H-1,2,3-triazol-4-ylmethoxy group at C6 and no substitution at C8; however, compound **94** presents a methylpiperazine group instead of a tert-butyl at position N9, which considerably increases its potency against *T. cruzi* (Figure 1). Interestingly, the methylpiperazine group decreased the activity of **94** against amastigote forms, yielding an IC_50_ value of 75.11 (6.4) µM (Table 2); this structure–activity relationship (SAR) was previously observed with compounds from group C. In agreement with previous results obtained with compounds **76** and **84**, the basic methylpiperazine group at N9 of compound **94** seemed to increase the toxicity against Vero cells (TC_50_ > 60 µM) compared to non-basic substituents, such as the tert-butyl substitution present in compound **88** (TC_50_ > 180 µM) (Table 2).

Finally, compound **93**, from chemical group E, was the second most potent compound of the collection in the anti-*T. cruzi* assay (IC_50_ = 1.42 (0.19) µM; Table 2). Similarly to compound **94**, when assessed against intracellular amastigotes, its activity considerably decreased (IC_50_ = 218.2 (69.1) µM; Table 2); it shared with compound **94** a methylpiperazine at N9 and no substitution at position C8, despite their being from different chemical groups (Figure 1). The 3-methoxyphenyl triazolyl methoxy group at position C6 of compound **94,** as opposed to the propargyloxy moiety present in compound **93,** seemed to increase its anti-*T. cruzi* activity. Interestingly, compound **29**, from subgroup C2, also shared a methylpiperazine at N9 and no substitution at C8 (Figure 1). Nevertheless, the presence of a benzoxy group at C6 could explain its lower anti-*T. cruzi* activity and selectivity (IC_50_ = 4.86 (0.65) µM, SI >205.8) in comparison to compound **93** (IC_50_ = 1.42 (0.19) µM, SI > 1408.45) (Table 2). It is also interesting to note that the toxicity to Vero cells of those three compounds increased as the substituent of C6 became more voluminous, although all reached high SI values (Table 2). Similarly, the potency against amastigote forms of those three compounds also increased following this substitution pattern (Table 2).

In vitro assays led to the identification of compounds **33** and **76** as hit compounds against *P. falciparum*, and compounds **6D** and **34** against *T. cruzi*. Since purine derivatives aim to target parasites´ purine salvage pathways, we performed an in silico docking study with available 3D structures or computationally generated models of the enzymes involved in these pathways. As natural ligands would be competing against those compounds for binding to the active site, we considered the differences in binding energy between compounds and natural ligands as the main parameter by which to elucidate potential targets. Thus, compounds **33** and **76** reported the highest differences in energy when docked with *Pf*HGXPRT (Table 3). Moreover, both compounds were bound with the highest affinity to that among all *P. falciparum* enzymes. Similar results were obtained for *Pf*ADA, suggesting that both enzymes could be potential targets. Regarding compounds **6D** and **34**, the highest energy difference was also shown against *Tc*HGPRT (Table 4). Similarly, *Tc*APRT reported considerable enough energy differences for both compounds to be considered a potential target (Table 4).

Visualization of *Pf*HGXPRT and *Tc*HGRT docking results revealed that compounds cover a more extensive space of the active site than the natural ligand, hypoxanthine (Figure 7 and Figure 8). Hypoxanthine re-docking with *Pf*HGXPRT correlates with the co-crystalized complex, and its binding with *Tc*HGPRT with previous reports [41]. In all cases, an aromatic group of each of the compounds prioritized was observed occupying the hydrophobic binding pocket of the hypoxanthine (Figure 7A and Figure 8A). Hydrophobic interactions are the main driving force in drug–receptor interactions, and the most frequent in high-efficiency ligands [42]. In general, the compounds showed more hydrophobic interactions than hypoxanthine. In *Pf*HGXPRT, interaction with I146 and a pi–pi stacking interaction with F197 were present in all bindings, while that involving Y116 was observed with compound **33** and hypoxanthine (Figure 7C). In *Tc*HGPRT, compounds **6D** and **34** shared with hypoxanthine a hydrophobic interaction with I113 and a pi–pi stacking interaction with F164 (Figure 8C). Hydrogen bonds are the second most common type of interaction between proteins and ligands [42]. Compounds formed fewer hydrogen bonds with HGPRT enzymes than did natural ligands, with the exception of compound **34**. None of the compounds shared hydrogen bonds with hypoxanthine. The frequency of hydrogen bonds was reduced from 59% to 34% of that of hydrophobic contacts in efficient ligands [42], which correlates with the types of interactions that the compounds formed, and with their high binding affinities. However, our docking study did not consider water molecules, or the flexibility of the residues of the active site. Hit compounds lack polar hydrogens, but present a high quantity of heteroatoms that might form water-mediated hydrogen bonds with other residues. 

All of the compounds showed a similar binding mode between them, and seemed to share fundamental hydrophobic interactions. An aromatic group of each compound located to the hypoxanthine binding site, while the rest of the molecule occupied the pyrophosphate, ribose, and 5′-phosphate binding sites of either *Pf*HGXPRT or *Tc*HGPRT. Compounds **33** and **76** showed a similar binding mode to those of previously described immucillin 5′-phosphates and acyclic nucleoside phosphonates [25,43,44]. Immucillin 5′-phosphates are transition state analogs of *Pf*HGXPRT, with *K*_i_ values between 1 and 4 nM [43], and several immucillins have undergone preclinical evaluation in malaria animal models [45,46]. Compounds within acyclic nucleoside phosphonates replace the labile phosphate group of immucillins with a phosphonate, avoiding the activity of phosphomonoesterases.

Regarding *Tc*HGPRT, compounds **6D** and **34** seem to bind similarly to two compounds of the collection described by Freyman et al. [18]. All of the compounds bind within the central region of the binding site, and share interactions with F164 and I113 [18]. Other studies targeting *Tc*HGPRT identified purine analogs with electronegative atoms bound to C6 of the purine ring [21], which is also the case with our collection. In compound **34**, electronegative atoms of the substitution at C6 of the purine ring resemble interactions of the 5′-phosphate group with loop III (residues 111 to 120) [18]. The N9 of the purine base is a key position in the HGPRT’s catalyzed reaction, since it is the site where the ribose of phosphoribosyl pyrophosphate (PRPP) is covalently linked to the purine base during nucleotide formation [47]. Consequently, the removal of this free amino group from the purine scaffold by the formation of *N*-substituted analogs would prevent the enzymatic incorporation of PRPP to the purine ring. This structural feature is present in our compound library, with the best candidates bearing, at position N9, a piperidin-1-yl in compound **76**, a tert-butyl in compound **6D,** and a *N,N*-dimethylamino group in compounds **33** and **34** (Figure 9). This structural feature might suggest that, if HGPRT is the target of the compounds, those would be acting as true inhibitors and not as alternate substrates [21]. In any case, confirmation of HGPRT as a potential target should be carried out via enzymatic assays and/or compound crystallization in complex with the enzyme. Such assays may also be performed with the *Pf*ADA and *Tc*APRT enzymes, considering the binding energies reported in the in silico docking study.

## 4. Materials and Methods

### 4.1. Chemical Collection

The compound library is comprised of 81 purine derivatives that belong to 5 different families (A–E) (Table 5). All of the compounds were synthesized and characterized by 1H-NMR, 13C-NMR, and HRMS. The synthesis and structural characterization data of purines from library groups A, D, and E were previously reported [29,48,49]. Structural characterization data (_1_H-NMR, ^13^C-NMR, and HRMS) from purine derivatives belonging to families B and C can be found in the Appendix A. Compounds were first screened at 1, 10, and 100 µM, and those progressed were assayed in a dose–response manner at starting concentrations of 100, 200, or 400 µM. The final DMSO percentage per well was in all cases below 0.5%.

### 4.2. Host Cell Cultures

Human erythrocytes were purified from B+ Rh+ blood samples and maintained at 4 °C in incomplete RPMI medium. Vero (green monkey kidney epithelial cells), LLC-MK2 (Rhesus monkey kidney epithelial cells), and HepG2 (human liver epithelial cells) cultures were cultivated in DMEM supplemented with 1% penicillin–streptomycin (PS) and 10% heat-inactivated fetal bovine serum (FBS) at 37 °C, 5% CO_2_, and >95% humidity, as described in [31]. HepG2 cells were also supplemented with 10% non-essential amino acids. Vero and MK2 cells were passaged twice per week at 1:12 ratios, while HepG2 cells were passaged at a 1:8 ratio.

### 4.3. Culture of P. falciparum and T. cruzi Parasites

*P. falciparum* 3D7 parasites were cultured with human erythrocytes in RPMI medium supplemented with 25 mM HEPES, 100 µM hypoxanthine, 25 mM bicarbonate, 5.5 mM glucose, 110 µM gentamicin, and 0.5 % AlbuMAX II. Apicomplexan parasite cultures were maintained at 3% hematocrit under an atmosphere of 37 °C in 93% N_2_, 5% CO_2_, and 2% O_2_, following standard methods [50]. Synchronized cultures were obtained via 5% sorbitol lysis [51].

*T. cruzi* parasites from the Tulahuen strain (discrete typing unit (DTU) VI) expressing β-galactosidase were maintained in culture by the infection of LLC-MK2 cells in DMEM supplemented with 2% FBS and 1% penicillin–streptomycin–glutamine (PSG), as described in [52]. When free-swimming trypomastigotes were released to the culture medium, they were centrifuged for 7 min at 2500 rpm and allowed to swim out of the pellet. Trypomastigotes were used to maintain the parasite cycle, or for the performance of the antiparasitic assays.

### 4.4. P. falciparum Primary Screening and Growth Inhibition Assay

*P. falciparum* growth was primarily analyzed under 3 different concentrations (100, 10, and 1 mM) of the 81 compounds. Conditions were tested in triplicate for every concentration, and the test was repeated when the standard deviation was above 10% for any concentration. *Z*-values varied between 0.83 and 0.98 in the different plates analyzed [53]. Selected compounds (12.3% potential hit rate) were further analyzed via standard growth inhibition assays, carried out as previously described in [54]. Briefly, parasitemia was adjusted to 1–1.3%, with >90% rings after sorbitol synchronization. Two hundred microliters of parasite culture were plated in 96-well microplates and incubated for 48 h at 37 °C in the presence of decreasing concentrations of the tested compounds, in triplicate. Parasitemia was determined by fluorescence-assisted cell sorting (FACS). Non-infected RBCs and samples containing parasitized RBCs (including controls with DMSO as carrier solvent) were diluted in PBS to a final concentration of ~1–10 × 10^6^ cells/mL. The cell suspension was stained with SYTO 11 (0.5 mM stock in DMSO, Thermo Fisher Scientific, Waltham, MA, USA), to a final concentration of 0.5 μM. Samples were analyzed in a Becton Dickinson FACSCalibur. Sample excitation was carried out using a 488-nm, air-cooled argon-ion laser at a power of 15 mW, using forward and side scatter to gate the RBC population, and SYTO 11 green fluorescence (530 nm, FITC filter) was collected in a logarithmic scale. The single-cell population was selected on a forward–side scattergram, and the green fluorescence from this population was analyzed. Parasitemia was expressed as the number of parasitized cells per 100 erythrocytes.

### 4.5. T. cruzi Primary Screening and Growth Inhibition Assay

*T. cruzi* growth was primarily analyzed under three different concentrations (100, 10, and 1 mM) of the 81 compounds. Conditions were tested in triplicate for every concentration. *Z*-values varied between 0.81 and 0.92 in the different plates analyzed [53]. Selected compounds (34.6% potential hit rate) were further analyzed via standard growth inhibition assays [31]. The *Z*-values of dose–response assays correlated with those previously reported [31]. For both assays, Vero cells and trypomastigotes from the Tulahuen *T. cruzi* strain expressing β-galactosidase [52] were harvested, centrifuged, and resuspended in DMEM without phenol red supplemented with 1% PSG, 2% FBS, 1 mM sodium pyruvate, and 25 mM HEPES. Vero cells and trypomastigotes were counted, diluted at a concentration of 1 × 10^6^ cells/mL each, and mixed 1:1 (*v*/*v*). Then, 100 μL of that mixture was added per well to the plates already containing the compounds. Each well contained 50,000 host cells and 50,000 parasites, i.e., multiplicity of infection (MOI) = 1. BNZ was used as control of maximal drug growth inhibition in each round, whereas each plate contained its own negative control (maximum parasite growth; Vero cells plus parasites without drugs) and positive control (minimum parasite growth; trypomastigote forms alone marking an enzymatic zero time or baseline galactosidase activity) [31]. After four days at 37 °C, 50 µL of a PBS solution containing 0.25% NP40 and 500 µM chlorophenol red-β-d-galactopyranoside (CPRG) substrate were added per well [32]. Plates were incubated at 37 °C for 4 h, and their absorbance read out at 590 nm using an Epoch Gene5 spectrophotometer.

### 4.6. Vero and HepG2 Toxicity Assays

Vero and HepG2 cells were detached, centrifuged, and resuspended in DMEM without phenol red. Cell viability was checked upon cell counting with trypan blue staining. Then, Vero and HepG2 cells were diluted at a concentration of 5 × 10^5^ and 3.2 × 10^5^ cells per mL, respectively, before adding 100 µL per well to the 96-well plate. Each run contained its own negative (untreated cells) and positive (medium alone) controls [31]. Plates were incubated at 37 °C for 4 days in the case of Vero, or 2 in the case of HepG2 cells. Assay readout was made by adding 50 µL per well of a PBS solution containing 10% alamarBlue reagent (Thermo Fisher Scientific); then, plates were incubated for another 6 h at 37 °C, before recording the fluorescence intensity with a Tecan Infinite M Nano+ reader (excitation: 530 nm, emission: 590 nm) [31].

### 4.7. Antiamastigote Specific Assay

Vero cells were seeded in T-175 flasks (5 × 10^6^ cells per flask) in DMEM supplemented with 1% penicillin–streptomycin and 10% FBS, and cultured for 24 h. Then, cells were washed once with PBS, and free-swimming trypomastigotes (1 × 10^7^ trypomastigotes per flask; MOI = 1) in assay medium were added and allowed 18 h to infect [55]. After that, infected cell monolayers were washed with PBS and detached. Cells were counted and diluted to a concentration of 5 × 10^5^ cells per mL, before adding 100 µL per well to test plates already containing the drugs dispensed as described above [55]. In all cases, we included BNZ as control drug, and each plate contained its own negative (Vero cells and parasites) and positive (Vero cells) controls [55].

### 4.8. Computational Analysis

A computational approach was followed to elucidate the mechanism behind the action of these compounds. A curated search of enzymes involved in the purine salvage pathway was performed in the two specialized databases of our organisms of interest from the Eukaryotic Pathogen, Vector, and Host Informatics Resource (VEuPathDB) [56]: PlasmoDB [57] for *P. falciparum*, and TriTrypDB [58] for *T. cruzi*. For *P. falciparum*, proteins selected consisted of ADA, ADSL, ADSS, GMP synthase (GMPS), HGXPRT, inosine-5′-monophosphate dehydrogenase (IMPDH), PNP, and nucleoside transporters 1 to 4 (NT1–NT4). For *T. cruzi*, the enzymes selected consisted of ADA, ADSL, ADSS, AK, adenine phosphoribosyltransferase (APRT), guanine deaminase (GD), HGPRT, IAGNH, IG-NH, inosine-5′-monophosphate dehydrogenase (IMPDH), and methylthioadenosine phosphorylase (MTAP). Two transporters were also selected: nucleoside transporter 1 (NT1) and nucleobase transporter 2 (NB2). Since the Tulahuen strain is not well annotated, all entries were obtained from the CL Brener Esmeraldo-like strain. Entries were obtained for the *P. falciparum* 3D7 strain. All of the selected proteins are summarized in Appendix A.

Selected proteins were queried for their NCBI [59] RefSeq entries, and these were blasted with BLASTP [60] against the human (taxid: 9606) non-redundant (nr) protein database, using default scoring parameters and an expect threshold of 20,000. The highest identity hit for each query, measured as the number of identical positions divided by the total length of the queried protein, was selected. Each selected protein was also blasted against the Protein Data Bank (PDB) database [61], using the PAM30 matrix with a word size of 2, no compositional adjustments, gap existence cost of 9, and extension cost of 1, with an expect threshold of 20,000. The highest identity hit was selected as described above (Appendix A). Selected PDB hits with an identity above 99% with the curated protein were directly used for docking simulations. Parasites’ proteins that did not have any PDB hits with an identity of at least 50% were discarded from further analysis.

PDB entries with an identity between 50% and 99% were submitted to comparative modelling with the MODELLER [62,63] Python package in order to generate 3D structures to be used in docking simulations. The methodology for this approach consisted of a two-step process: first, an alignment of the RefSeq entry with the sequence of the PDB record, and second, the generation of five 3D models with MODELLER’s AutoModel class for each protein, using the previous alignment and the atom coordinates from the PDB record as a template. The best model for each round was selected according to the highest GA341 [64] and lowest DOPE [65] scores. While the DOPE score can only be used to select the best structure from a collection of models built by the program for the same protein, the GA341 score can be used to generally assess the reliability of any given model. All of the models generated for this work scored 1 for this method—the highest possible value, and comparable with low-resolution structures [64].

Both high-identity PDB entries and computationally generated models were prepared for docking simulations. Water molecules, if present, were removed, and Kollman charges and polar hydrogens were added using AutoDockTools 1.5.6 [66]. Resulting structures were formatted as PDBQT files. PDB entries with co-crystallized cofactors were processed with and without them, generating two versions for each entry. 3D structures for the selected compounds were generated with Avogadro 1.2.0 [13], and their molecular geometry was optimized until the lowest energy values were obtained. Natural ligands’ 3D structures were obtained from PubChem [67] as SDF files, as follows: adenine (PubChem ID 190), adenosine (PubChem ID 60961), adenylosuccinic acid (PubChem ID440122), guanine (PubChem ID 135398634), guanosine (PubChem ID 135398635), hypoxanthine (PubChem ID 135398638), inosine (PubChem ID 135398641), inosine monophosphate (IMP) (PubChem ID 135398640), SAICAR (PubChem ID 160666), xanthine (PubChem ID 1188), and xanthosine monophosphate (XMP) (PubChem ID 73323). PDBQT files of the ligands were also generated with AutoDockTools, with all the default values accepted.

Docking was performed with AutoDock Vina 1.1.2 [68]. The binding box used was centered on the active site of each enzyme, using data from the PDB and/or literature (Appendix A); ligands’ binding boxes were constrained to any previously described position, if possible. Exhaustiveness was set to 12, and energy range to 4. Each round of docking with AutoDock Vina produced 9 different binding modes, from which the mode with the lowest binding energy (in Kcal/mol) was selected. This was repeated 100 times for each enzyme’s natural ligand(s) and compounds, using a different random seed each round; in structures with cofactors, docking simulations were repeated with and without these present. Means and standard deviations for the binding energies were calculated with the values from the selected 100 best binding modes. LigPlot+ v.2.2.4 [69] was used to analyze different types of protein–ligand interactions with default parameters, and PyMOL 2.4.1 [70] was used to visualize and image the results. A hydrophobicity surface map was generated for selected enzymes using the Kyte–Doolittle hydrophobicity scale [71], and the electrostatic potential map was generated using PyMOL’s APBS Electrostatics plugin [72].

### 4.9. Data Analysis

Absorbance values derived from the anti-*T. cruzi* assays and fluorescence values from the anti-*P. falciparum* and cell toxicity assays were normalized to the controls [33]. In the primary screening at three fixed doses, the compounds’ inhibitory activity was expressed as a percentage of the negative control (100% parasitic growth). In dose–response assays, the compounds’ IC_50_ and TC_50_ values were determined with GraphPad Prism 7 software (version 7.00, 2016), using a non-linear regression analysis model defined by the following equation:$$Y=100\;÷\left(1+10^{\left(\left(LogIC50-X\right)\times HillSlope\right)}\right)$$

IC_50_ and TC_50_ values are respectively provided as means and standard deviations (SD), or as the minimum value of the 95% confidence interval (CI) of at least three experiments.

## 5. Conclusions

In this study, we analyzed the antiparasitic activity of a library of 81 purine and pyrimidine analogs by screening them against *P. falciparum* and *T. cruzi*. Around 50% of these molecules showed some biological activity by inhibiting parasites’ growth, and the majority of them had low toxicity to Vero and HepG2 cells. The most potent hits against *P. falciparum* were purines **33** and **76,** with IC_50_ < 20 µM, both sharing an N-substituted triazolylmethoxy group at C6, a tertiary amine at N9, and a phenyl group at N8 (Figure 9). On the other hand, compounds **6D** and **34** were the most active compounds against intracellular amastigotes of *T. cruzi*, with IC_50_ values of 3.78 (0.31) and 4.24 (0.29) µM, respectively. Interestingly, both of these compounds contain a phenyl group at C8 and a small substituent at N9, which seems to be crucial for displaying activity against amastigote forms (Figure 9). Our docking study further suggested that *Pf*HGXPRT and *Tc*HGPRT enzymes could be the main potential targets of those compounds. Thus, two novel, purine-based chemotypes with potent activities against either *P. falciparum* or *T. cruzi* have been identified, and could be further optimized to eventually generate potent and diverse antiparasitic drugs.

## Figures and Tables

**Figure 1 pharmaceuticals-14-00638-f001:**
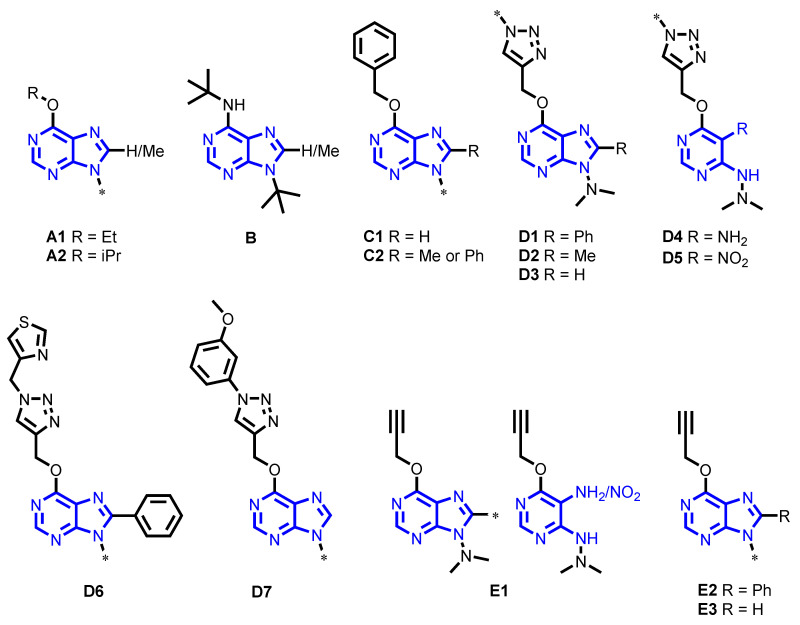
Chemical structures and classifications of the evaluated compounds. Letters A–E identify the five different chemical groups (scaffolds), while their accompanying numbers depict scaffold subgroups (see Section 4.1 for further details). *, indicates different chemical substituents.

**Figure 2 pharmaceuticals-14-00638-f002:**
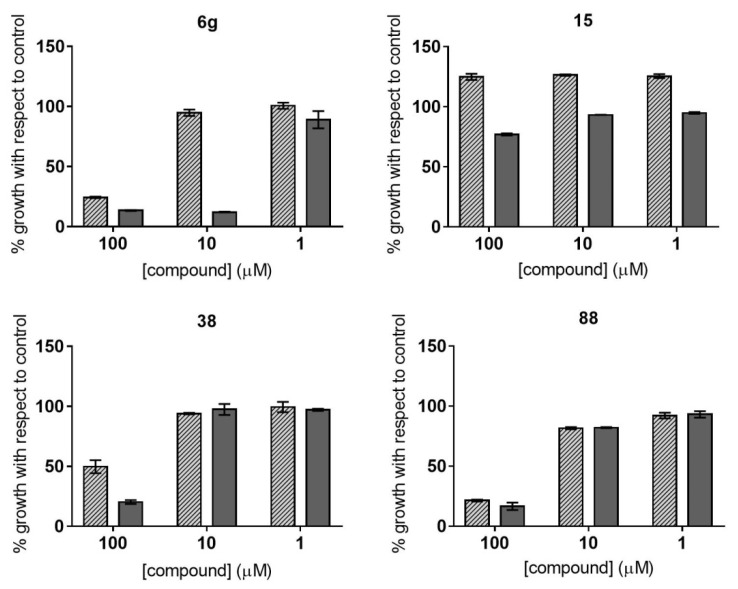
Parasite growth at three concentrations of representative compounds from different structural groups. Filled bars represent *P. falciparum* growth, whereas dark grey bars represent *T. cruzi* growth.

**Figure 3 pharmaceuticals-14-00638-f003:**
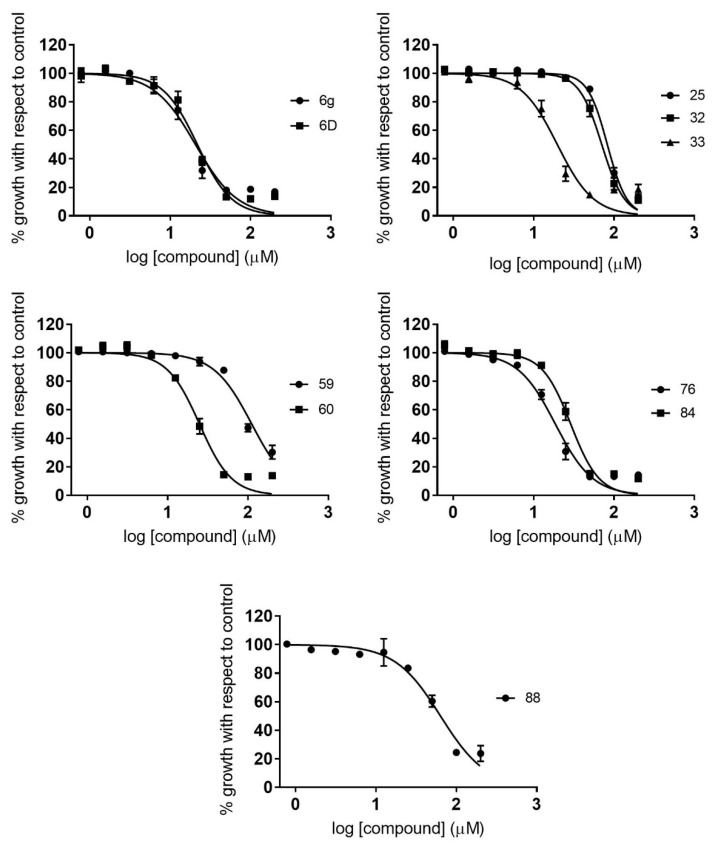
Dose–response curves of active compounds against *P. falciparum*, classified according to different structural groups (from A to E). Graphs represent mean values and SD results of at least three replicas.

**Figure 4 pharmaceuticals-14-00638-f004:**
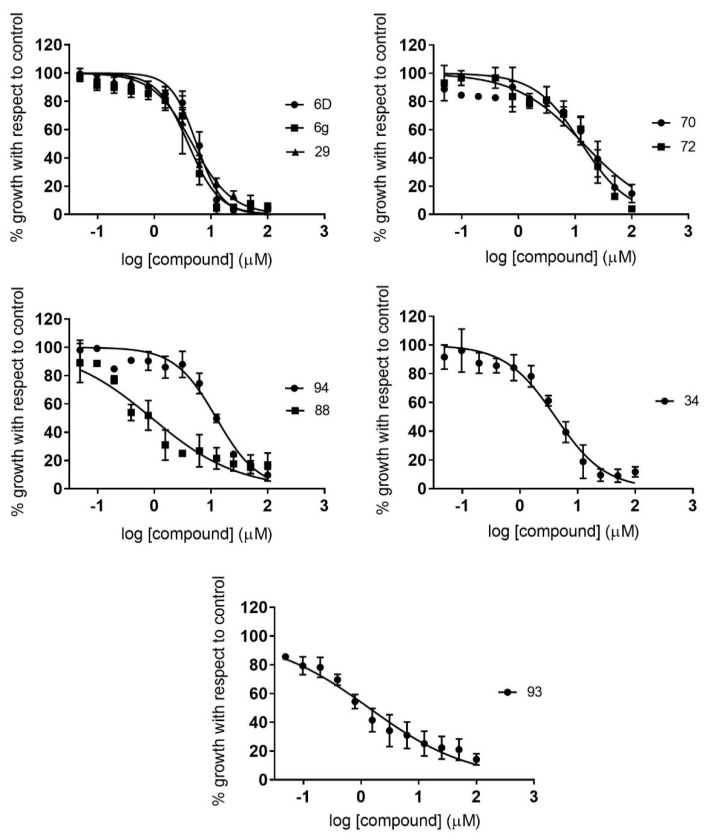
Dose–response curves of active compounds against *T. cruzi*, classified according to the different structural groups (from A to E). Graphs represent mean values and SD results of at least three independent replicas.

**Figure 5 pharmaceuticals-14-00638-f005:**
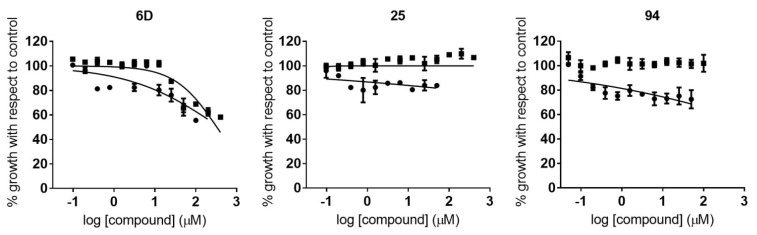
Dose–response curves of some of the active compounds against *P. falciparum* and *T. cruzi* in Vero and HepG2 cell toxicity assays. Vero cell toxicity assays are represented by circles, whereas HepG2 cell toxicity assays are represented by squares. Graphs represent mean values and SD results of at least three independent replicas.

**Figure 6 pharmaceuticals-14-00638-f006:**
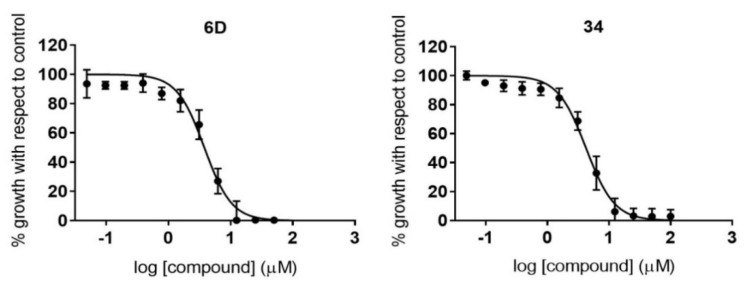
Dose–response curves of compounds with specific antiamastigote activity. Graphs represent mean values and SD results of at least three independent replicas.

**Figure 7 pharmaceuticals-14-00638-f007:**
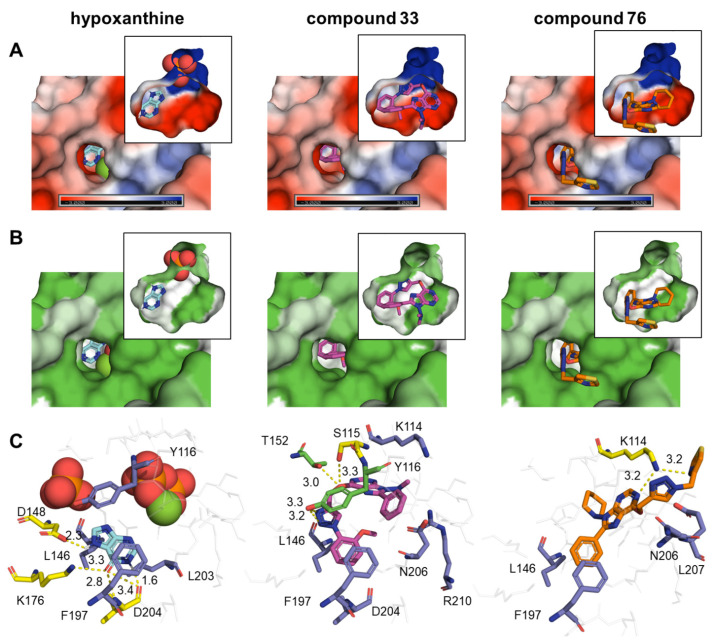
Binding of natural ligand and hit compounds with *Pf*HGXPRT. (**A**) Electrostatic surface view of the binding pocket of *Pf*HGXPRT. (**B**) Hydrophobicity surface view of the binding pocket of *Pf*HGXPRT. (**C**) Zoomed-in view of the interactions between *Pf*HGXPRT active site residues and their corresponding natural ligands or hit compounds. In panel C, residues that form hydrophobic interactions at a maximum distance of 3.9 Å are shown as purple sticks, those that formed H bonds as yellow sticks, and those involved in both types of interactions as green sticks, while non-interacting residues are represented as grey sticks. Expected H bonds are represented in yellow. In panel B, the hydrophobicity surface map was generated using the Kyte–Doolittle hydrophobicity scale. Red and orange spheres represent a pyrophosphate and a phosphate group, whereas the green sphere is a magnesium group.

**Figure 8 pharmaceuticals-14-00638-f008:**
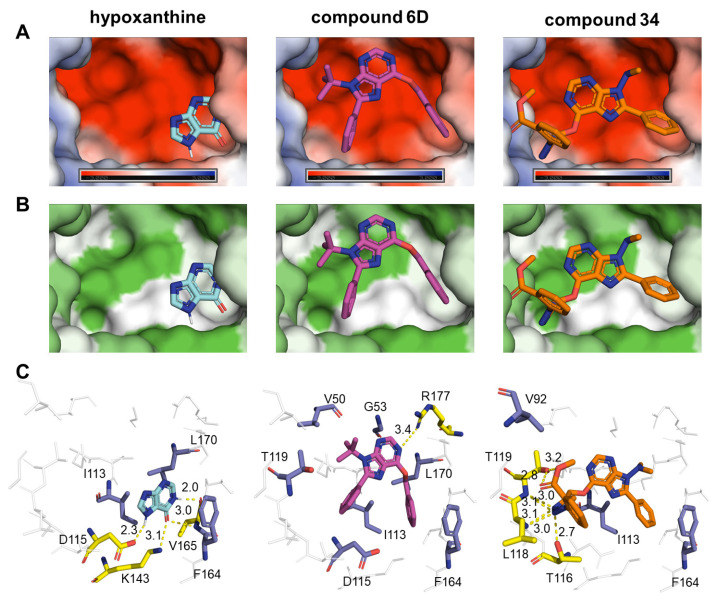
Binding of natural ligand and hit compounds with *Tc*HGPRT. (**A**) Electrostatic surface view of the binding pocket of *Tc*HGPRT. (**B**) Hydrophobicity surface view of the binding pocket of *Tc*HGPRT. (**C**) Zoomed-in view of the interactions between HGPRT active site residues and their corresponding natural ligands or hit compounds. In panel C, residues that form hydrophobic interactions at a maximum distance of 3.9 Å are shown as purple sticks, and those that form H bonds as yellow sticks, while non-interacting residues are grey sticks. Expected H bonds are represented in yellow. In panel B, the hydrophobicity surface map was generated using the Kyte–Doolittle hydrophobicity scale.

**Figure 9 pharmaceuticals-14-00638-f009:**
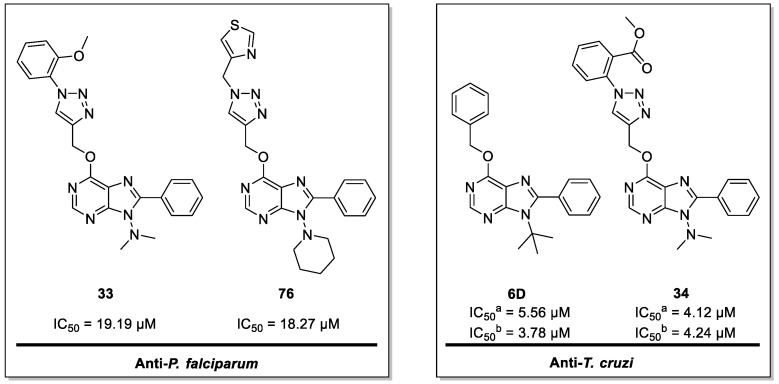
Chemical structures of the lead compounds against *P. falciparum* and *T. cruzi*. ^a^: anti-*T. cruzi* primary assay; ^b^: antiamastigote assay.

**Table 1 pharmaceuticals-14-00638-t001:** Number of compounds classified according to the percentage of parasite growth in the primary screening.

	Parasite Growth Rate Relative to the Assay Negative Control					
Compound Concentration (µM)	*P. falciparum*			*T. cruzi*		
	<30%	30–50%	>50%	<30%	30–50%	>50%
**100**	10 (12.3) *	10 (12.3)	61 (75.3)	28 (33.3)	8 (9.8)	45 (56.8)
**10**	1 (1.2)	0 (0)	80 (98.8)	2 (2.4)	2 (2.4)	77 (95.0)
**1**	0 (0)	0 (0)	81 (100)	0 (0)	0 (0)	81 (100)

*: percentage of compounds out of the 81 screened in total is shown in parentheses.

**Table 2 pharmaceuticals-14-00638-t002:** IC_50_ (μM), TC_50_ (μM), and SI values of active compounds against *P. falciparum* and *T. cruzi*.

		*P. falciparum*	*T. cruzi*		Vero Cells				HepG2 Cells
Compound	Subgroup	IC_50_	IC_50_ ^a^	IC_50_ ^b^	TC_50_ *	SI ^c^	SI ^a^	SI ^b^	TC_50_ *
**BNZ**	-	NT	1.72	1.82	209.5	-	121.80	115.11	255.3
**29**	C1	NT	4.86	157.3	>1000	-	>205.76	>6.36	>1000
**6D**	C2	18.9	5.56	3.78	>150	>7.94	>26.98	>39.68	>200
**6g**	C2	19.08	3.97	19.64	>150	>7.86	>37.78	>7.63	>300
**25**	D1	81.54	NT	NT	>6000	>73.58	-	-	NA
**32**	D1	69.66	NT	NT	>1000	>14.36	-	-	NA
**33**	D1	19.19	NT	NT	>300	>15.63	-	-	NA
**34**	D1	NT	4.12	4.24	>6000	-	>1456.31	>1415.10	NA
**59**	D5	110.1	NT	NT	>300	>2.73	-	-	NT
**60**	D5	23.3	NT	NT	>300	>12.87	-	-	NA
**70**	D6	NT	13.72	29.08	>50	-	>3.64	>1.72	NT
**76**	D6	18.27	NT	NT	>500	>27.37	-	-	NA
**84**	D6	28.01	NT	NT	>70	>2.50	-	-	NT
**88**	D7	65.35	12.72	35.43	>180	>2.75	>14.15	>5.08	NA
**94**	D7	NT	0.92	75.11	>60	-	>63.16	>0.79	NA
**93**	E3	NT	1.42	218.2	>2000	-	>1408.45	>9.17	NA

^a^: anti-*T. cruzi* assay; ^b^: antiamastigote assay; ^c^: anti-*P. falciparum* assay. *: Values are expressed as the minimum value of the 95% CI, except for those of BNZ. NT: not tested; NA: not adjusted.

**Table 3 pharmaceuticals-14-00638-t003:** Free binding energies of molecular docking between natural ligands and hit compounds with enzymes from the purine salvage pathway of *P. falciparum* 3D7.

Enzyme	Natural Ligand	Energy Binding with Natural Ligand (Kcal/mol)	Energy Binding with Compound 33 (Kcal/mol)	Energy Binding with Compound 76 (Kcal/mol)	Energy Difference with 33 (Kcal/mol)	Energy Difference with 76 (Kcal/mol)
PfHGXPRT (XP_001347406.1)	Xanthine	−6.80 (0.00)	−9.91 (0.08)	−10.30 (0.02)	−3.11	−3.50
	Hypoxanthine	−5.73 (0.26)			−4.18	−4.58
	Guanine	−6.65 (0.05)			−3.26	−3.65
PfADA (XP_001347573.1)	Adenosine	−7.10 (0.00)	−9.05 (0.05)	−9.22 (0.09)	−2.12	−1.95
PfGMPS (XP_001347408.1)	XMP	−6.63 (0.12)	−8.19 (0.19)	−8.51 (0.06)	−1.56	−1.88
PfPNP (XP_001351690.1)	Inosine	−7.13 (0.04)	−8.51 (0.14)	−8.83 (0.46)	−1.38	−1.71
PfADSL (XP_001349577.1)	SAICAR	−8.10 (0.13)	−9.27 (0.11)	−9.15 (0.06)	−1.17	−1.05
	Adenylosuccinate	−8.41 (0.12)			−0.86	−0.74
PfADSS (XP_001350257.1)	IMP	−9.49 (0.03)	−9.62 (1.02)	−8.51 (0.31)	−0.13	0.98

**Table 4 pharmaceuticals-14-00638-t004:** Free binding energies of molecular docking between natural ligands and hit compounds with enzymes from the purine salvage pathway of *T. cruzi* CL Brenner.

Enzyme	Natural Ligand	Energy Binding with Natural Ligand (Kcal/mol)	Energy Binding with Compound 6D (Kcal/mol)	Energy Binding with Compound 34 (Kcal/mol)	Energy Difference with 6D (Kcal/mol)	Energy Difference with 34 (Kcal/mol)
TcHGPRT (XP_813396.1)	Hypoxanthine	−4.80 0.00)	−7.92 (0.04)	−8.64 (0.13)	−3.12	−3.84
	Guanine	−5.49 (0.03)			−2.43	−3.15
TcAPRT (XP_818435.1)	Adenine	−5.40 (0.02)	−7.64 (0.34)	−8.37 (0.36)	−2.23	−2.97
TcAK (XP_820251.1)	Adenosine	−7.24 (0.10)	−9.20 (0.00)	−9.72 (0.09)	−1.96	−2.48
TcIGNH (XP_818171.1)	Inosine	−7.62 (0.09)	−9.10 (0.02)	−9.25 (0.11)	−1.48	−1.63
	Guanosine	−8.22 (0.20)			−0.88	−1.03
TcIMPDH (XP_805772.1)	IMP	−7.99 (0.03)	−8.19 (0.04)	−9.12 (0.24)	−0.20	−1.13
TcIAGNH (XP_804829.1)	Inosine	−9.40 (0.00)	−8.92 (0.13)	−10.09 (0.17)	0.48	−0.69
	Guanosine	−9.33 (0.27)			0.41	−0.76
	Adenosine	−8.85 (0.05)			−0.07	−1.21
TcADSL (XP_811726.1)	SAICAR	−9.26 (0.08)	−8.50 (0.00)	−10.08 (0.32)	0.76	−0.82
	Adenylosuccinate	−9.45 (0.06)			0.95	−0.63

Sequence ID references at NCBI are shown for each enzyme. Energy difference value is calculated as the difference between the binding energy of the compound and that of the natural ligand.

**Table 5 pharmaceuticals-14-00638-t005:** Classification of the compounds within the chemical collection evaluated in this study.

Group	Scaffold	Sub-Group	Chemical Name	Number of Compounds
A	6-Alkoxy-purines	A1	6-Ethoxy-8-(*H* or methyl)-9-substituted-9*H*-purines	4
		A2	8-(*H* or Methyl)-6-isopropoxy-9-isopropyl-9*H*-purines	2
B	6-*Tert*-butyl amino purines	-	9-*tert*-Butyl-6-(*tert*-butylamino)-8-substituted-9*H*-purines	3
C	6-Benzoxy-purines	C1	6-Benzoxy-9-substituted-9*H*-purines	4
		C2	6-Benzoxy-9-substituted-8-(methyl or phenyl)-9*H*-purines	10
D	6-{[1-(substituted)-1*H*-1,2,3-triazol-4-yl]methoxy}-purines and pyrimidine analogs	D1	6-{[1-(substituted)-1*H*-1,2,3-triazol-4-yl]methoxy}-9-(dimethylamino)-8-phenyl-9*H*-purines	7
		D2	8-Methyl-9-(dimethylamino)-6-{[1-(substituted)-1*H*-1,2,3-triazol-4-yl]methoxy}-9*H*-purines.	7
		D3	6-{[1-(substituted)-1*H*-1,2,3-triazol-4-yl]methoxy}-9-(dimethylamino)-9H-purines	7
		D4	6-{[1-(substituted)-1*H*-1,2,3-triazol-4-yl]methoxy}-4-(2,2-dimethylhydrazinyl)-5-nitropyrimidines	7
		D5	5-Amino-4-(2,2-dimethylhydrazinyl)-6-{[1-(substituted)-1*H*-1,2,3-triazol-4-yl]methoxy}pyrimidines	7
		D6	9-(Alkyl or Dialkylamino)-8-phenyl-6-{[1-(thiazol-4-ylmethyl)-1*H*-1,2,3-triazol-4-yl]methoxy}-9*H*-purines	5
		D7	9-(Alkyl or Dialkylamino)-6-{[1-(3-methoxyphenyl)-1*H*-1,2,3-triazol-4-yl]methoxy}-9*H*-purines	5
E	6-Propargyloxy-purines and pyrimidine analogs	E1	9-(*N*,*N*-dimethyl)-6-propargyloxy-8-substituted-9*H*-purines and pyrimidine analogs	5
		E2	9-(Alkyl or Dialkylamino)-8-phenyl-6-propargyloxy-9*H*-purines	4
		E3	9-(Alkyl or Dialkylamino)-6-propargyloxy-9*H*-purines	4

## Data Availability

The data presented in this study are available in the article or Appendix A.

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
