# Peer review of "Novel Purine Chemotypes with Activity against Plasmodium falciparum and Trypanosoma cruzi"

_pharmaceuticals, 2021, doi:10.3390/ph14070638_

Round 1

Reviewer 1 Report

Thanks to the authors for their reply to all the points raised by reviewer 3. However, based on the modified manuscript, there are some concerns that the authors should address before this manuscript would be suitable for publication;

1- the authors should submit the NMR spectra for All compounds in SI where the spectra presented H-NMR (0-12ppm) and C-NMR (0-210ppm). All the spectra supported as cut/focused area only. Please, be aware that the spectra for compds 3F, 2A and 1B are not visible.

2- the authors did not modify the intro as was suggested in the first report. (authors should mention something about the purine salvage pathways of parasites, what could be the target of purine analogues.)

3- Regarding the performed in silico study, the authors have discussed and presented their results based only on the binding energy, which must be supported by the binding mode and the type of interactions. Indeed, fig7 is not informative in the presented form. The authors should present their results by discussing the type of interactions (hydrophilic and hydrophobic) comparing this to the co-crystallized ligand. Further, what was the results of re-dock of co-crstallize ligand? Were all essential bindings detected? Did the tested compounds show similar binding mode as co-crstallise ligand, or they could form new binding mode? All such points should be addressed.

Author Response

We thank the reviewer for his/her comments, which have greatly contributed to improve the manuscript.

Answers to the three points raised are pasted below:

1. The supporting information has been update to include additional compounds. Moreover, all the 1H-NMR and 13C-NMR spectra have been modified to be presented as focused area only, and spectra for compounds 3F, 2A and 1B have been corrected too and are now visible.

2. Introduction section has been now modified to incorporate information about the purine salvage path of both parasites and the enzymes that have been studied as targets of drug discovery efforts (see page 2 of 28 within the revised manuscript).

3. These features have now been addressed in the resubmitted new version of the manuscript. For this, new text has been incorporated in Results section 2.6, including revised Tables 3 and 4, as well as new Figures 7 and 8. Changes have also been made accordingly in the Discussion and Methods sections where needed. 

Reviewer 2 Report

The authors have addressed my main crticism and, as a result, they have significantly improved the quality and the impact of their manuscript.

In my opinion, the manuscript can now be published in Pharmaceutics.

Author Response

Thank you for your time and dedication.

Round 2

Reviewer 1 Report

Thanks to the authors for addressing and clarifying all concerns raised. The manuscript has been significantly modified, and therefore I would recommend the publication of this interesting study.

This manuscript is a resubmission of an earlier submission. The following is a list of the peer review reports and author responses from that submission.

Round 1

Reviewer 1 Report

In this manuscript, Martinez-Peinado et al. have analyzed the in vitro anti-P. falciparum and anti-T. cruzi activity of a collection of 81 purine derivatives and pyrimidine analogs. Their study revealed novel purine-based chemotypes that, albeit through unidentified mechanisms, appear to be active against both parasites at micromolar concentrations. Based on their structural features, the authors claim that some guidelines can be derived to guide further optimization with the aim to generate potent and diversified anti-parasitic drugs.

Overall, I think that this study is well conceived and well executed. Also, the manuscript is clearly written and provides good background information as well as extensive experimental details. Moreover, it highlights the need to adopt comprehensive approaches to exploit every possible strategy against deadly parasites.

However, what I think is missing (and I would have appreciated) in this manuscript is an attempt to elaborate on the actual mechanism behind these phenomenological observations. What are the most likely enzymes involved? Have any of these possible enzymes been structurally characterized? If so, would the authors conduct at least a quick computational analysis involving docking and molecular dynamics simulations to suggest (or even exclude) possible inhibitory mechanism? After all, the good potency of these inhibitors as well as the significant differences between members of the same subgroups seem to suggest that a specific inhibitory mechanism is taking place.

Overall, I think that the manuscript would benefit from a major revision whereby the authors can go a little beyond their phenomenological observations and provide an attempt to elaborate on possible inhibitory mechanisms. It would be difficult to make use of their interesting guidelines otherwise.

Author Response

We thank R1 for these comments and suggestions. Following them, in the re-submitted version of the manuscript we provide specific sections in Methods and Results about the potential targets of the prioritized compounds. These entail a computational-assisted new docking study. We further discuss these new results in the Discussion section, and finally, possible mechanisms of inhibition of the potential target enzymes have been now included in the text.

Reviewer 2 Report

The authors performed phenotypic screening of Plasmodium falciparum and Trypanosoma cruzi against 81 purine derivatives and pyrimidine analogs. They identified two purines that show anti-P. falciparum activity and another two purines show anti-T. curzi activity. Unfortunately, I can not recommend this manuscript to be accepted to Pharmaceuticals for the following reasons.

 -Authors argued that this work is a rational drug design because, unlike the human host, the parasites are incapable of de novo purine biosynthesis. But performing a phenotypic screening against a small set of small molecules is not considered a rational design of drug discovery (rational drug design refers to developing new drugs based on the knowledge of a specific molecular target, but not biosynthetic pathways).

-Screening anti-Plasmodium or anti-T. cruzi compounds against purine and pyrimidine derivatives have been done, and this work is not novel at all. If the authors want to identify inhibitors against specific targets (line 81 to 85), they should run target-based screenings using a specific target (protein) of interest.

-The screening hits (IC50 values) are not very potent at all (~20 µM). These days, the cut-off line for screening hits is below one µM or even lower.

-The authors failed to report the quality of the screening (for example, z’-factor and hit rates) and the details of screening hits (SAR table, for example).

Author Response

The authors performed phenotypic screening of Plasmodium falciparum and Trypanosoma cruzi against 81 purine derivatives and pyrimidine analogs. They identified two purines that show anti-P. falciparum activity and another two purines show anti-T. curzi activity. Unfortunately, I cannot recommend this manuscript to be accepted to Pharmaceuticals for the following reasons.

 - Authors argued that this work is a rational drug design because, unlike the human host, the parasites are incapable of de novo purine biosynthesis. But performing a phenotypic screening against a small set of small molecules is not considered a rational design of drug discovery (rational drug design refers to developing new drugs based on the knowledge of a specific molecular target, but not biosynthetic pathways).

We agree with the reviewer about the meaning of ‘rational drug design’. By bringing up the concept our aim was to pinpoint that we were focusing in much diverging metabolic routes between the host and the pathogen, as a practical starting point to potentially identify new hit compounds. Thus, we thank this reviewer for highlighting this inconsistency and we have changed accordingly the references to ‘rational drug design’ in the main text.

Page 1, line 28, Abstract: ‘... The discovery of new drugs may be benefited by considering significant biological differences between host and parasites ...’

Page 2, line 64, Introduction: ‘... Exploiting biochemical and physiological differences between parasites and hosts could contribute to the uncovering of new drugs. ...’

- Screening anti-Plasmodium or anti-T. cruzi compounds against purine and pyrimidine derivatives have been done, and this work is not novel at all. If the authors want to identify inhibitors against specific targets (line 81 to 85), they should run target-based screenings using a specific target (protein) of interest.

Actually, the novelty is the screening of a new collection of compounds, substituted purine derivatives and pyrimidine analogs, synthesized by generating variation in C6, C8 and C9 of the purine ring. In addition to more classical and highly lipophilic purine analogs (scaffolds A and B), most of the compounds of the library present two important structural novelties that have not been properly explored neither chemically nor in terms of their anti-parasitic activity and are therefore of great interest: the first is the presence of substituted 1,2,3-triazole groups attached at the C6 position of the purine or equivalent position for pyrimidine-derived analogs (scaffold D); and the second is the presence of dialkylamino groups attached at the purine N9 position or equivalent position for pyrimidine derivatives (scaffolds C, D and E), giving rise to endocyclic-exocyclic N-N bonds (examples: compounds 33, 34, 76). In our opinion, the results obtained from this phenotypical screening, especially those related to T. cruzi, may be of interest for those investigating the specific growth inhibition of protozoan parasites. Moreover, considering the different chemotypes of the molecules in the chemical collection evaluated in this study, the results may inform the improvement of purine-based compounds acting on the purine metabolism. Potential targeted-based screening might be implemented in the future using these or related molecules, but this is not at all part of the present research work.

- The screening hits (IC50 values) are not very potent at all (~20 µM). These days, the cut-off line for screening hits is below one µM or even lower.

We agree that the hits are not very potent for P. falciparum, where drugs currently in use show IC50 values at the low nM level in non-resistant parasitic strains. However, as stated above, the different chemotypes investigated could be informative for further exploration of compounds affecting the purine salvaging pathway. On the other hand, the most potent compounds against T. cruzi (i.e., 34 and 6D) are comparable to benznidazole, the first-line treatment against Chagas disease in most countries, and may represent a new starting point for the design of new compounds against this parasite. Furthermore, compound 34 presents a better profile of selectivity than benznidazole, a drug known by its poor tolerability profile and high incidence of treatment discontinuations.

- The authors failed to report the quality of the screening (for example, z’-factor and hit rates) and the details of screening hits (SAR table, for example).

-values of anti-T. cruzi and anti-P. falciparum primary screening assays are now provided in the text in sections 4.4 and 4.5. Moreover, the three different concentrations at 100, 10 and 1 mM were assayed in triplicate for each compound, and repeated when SD was higher than 10% at any different concentration.

Regarding the report of the screening hits, we have discussed the SAR of the most active compounds and compared the chemical structures of these hits with closely related inactive analogs in the results section. In addition to that, we have highlighted the chemical structures of the most active molecules. However, due to the high structural heterogeneity of the studied compound library, we believe that presenting these results in a SAR table format will provide very limited useful information to the reader.

Reviewer 3 Report

Alonso-Padilla et al reported the screening of 81 derivatives of purine-based molecules as anti-parasitic agents. Initially, the screening was performed at three different concentrations /100uM, 10uM, 1uM) and the compounds was filtered based on their inhibitory potency at 100uM (to kept the growth of the parasites < 30%). The screening led to discover of a small set of compounds which was subjected to two different cytotoxicity assays on Vero cells and human HepG2 34 cells. Finally, the most active compounds were tested against intracellular amastigote forms which led to discovery of Purines 33 (IC50 = 19.19 μM) and 76 (IC50 = 18.27 μM) as the most potent compounds against P. falciparum and compound 6D (IC50 = 3.78 μM) and 34 (IC50 = 4.24 μM) as hit compounds against T. cruzi amastigotes.

This work is interesting but lack of novelty and innovation. The activity of discovered compounds is not that potent and there are much potent purine analogues have been discovered earlier. Therefore, I do not see any scientific significance for this study. Especially, the tested 81 purine analogues were already available and no chemistry has performed. Additionally, the biological assays that have been performed (at very high concentrations) are very simple and non informative. It would be more attractive if the authors have tested the activity against QC-sensitive and resistant strains. Or, performing an in silico docking study to speculate the target of this class of compounds. Overall, I'm not supporting the publication of this study in the present form, unless additional detailed work included. The following is some suggestions;

the introduction part is not informative. very little about purine analogues and the relevant studies. Many studies have been reported and many purine derivatives are known, even over 32 patents. nothing is mentioned which giving impression to the reader that you are reporting something really novel. (https://doi.org/10.1517/13543776.15.8.987, DOI: 10.1039/c8md00098k, https://doi.org/10.1021/acs.jmedchem.8b00999, https://doi.org/10.1515/ap-2017-0070, https://www.sciencedirect.com/topics/biochemistry-genetics-and-molecular-biology/purine-analogue). Additionally, you should mention something about the the purine salvage pathways of parasites what could be the target of purine analogues.

the originality of the 81 compounds is not identified. Was it commercial or synthesized? if commercial please mention it, if synthesized, please cite the synthetic paper or add the NMR-spectra for the 81 compounds which prove the chracterization of these compounds.

it would be interesting to evaluate the activity of best compounds against resistant-strains which would make these compounds more attractive.

The authors should perform a docking study to show what the target of these compounds and what is the binding affainity; could be N-hydrolases (NH), purine nucleoside phosphorylase (PNP) and also 5′-deoxy-5′-methylthioadenosine phosphorylase (MTAP).

Author Response

This work is interesting but lack of novelty and innovation. The activity of discovered compounds is not that potent and there are much potent purine analogues have been discovered earlier. Therefore, I do not see any scientific significance for this study. Especially, the tested 81 purine analogues were already available and no chemistry has performed. We thank R3 for his/her comments and suggestions.

In order to respond to this first concern, we would like to emphasize that the compound library is comprised by 81 purine derivatives that belong to five different families (A-E). All presented compounds have been synthesized in our laboratory and are fully characterized by 1H-NMR, 13C-NMR and HRMS. Specifically, the synthesis and structural characterization data of purines from library A can be found here https://pubs.rsc.org/no/content/articlehtml/2015/ob/c5ob00230c , while for purine libraries D and E this information is available here https://chemrxiv.org/articles/preprint/Design_and_Synthesis_of_9-Dialkylamino-6-_1H-1_2_3-triazol-4-yl_methoxy_-9H-purines/13664516/1 . Lastly, as purine derivatives belonging to the families B and C have not been previously reported, we have now added their structural characterization data to the manuscript as supplementary material. The corresponding references have been included into the manuscript text to clarify the originality of the compounds. Please see “Supporting information” document as part of the manuscript supplementary material.

Additionally, the biological assays that have been performed (at very high concentrations) are very simple and non-informative. It would be more attractive if the authors have tested the activity against QC-sensitive and resistant strains.

We are afraid we do not have such strains available at the lab, and therefore cannot perform the suggested experiments with them.

Or performing an in silico docking study to speculate the target of this class of compounds.

We have now included in the re-submitted manuscript version a thorough computational analysis of the docking of the four prioritized compounds to those enzymes that could be their potential targets within the pathway. Specific text has been incorporated in Methods, Results and Discussion sections to explain it and contextualize it.   

Overall, I'm not supporting the publication of this study in the present form, unless additional detailed work included. The following is some suggestions:

- The introduction part is not informative. Very little about purine analogues and the relevant studies. Many studies have been reported and many purine derivatives are known, even over 32 patents. Nothing is mentioned which giving impression to the reader that you are reporting something really novel. (https://doi.org/10.1517/13543776.15.8.987, DOI: 10.1039/c8md00098k, https://doi.org/10.1021/acs.jmedchem.8b00999, https://doi.org/10.1515/ap-2017-0070, https://www.sciencedirect.com/topics/biochemistry-genetics-and-molecular-biology/purine-analogue). Additionally, you should mention something about the purine salvage pathways of parasites what could be the target of purine analogues.

We thank the reviewer for highlighting these references on the subject. Introduction section has been now modified to incorporate them. Also, information is now provided about the purine salvage path of parasites and the enzymes that could be potential targets of the compounds under study.

- The originality of the 81 compounds is not identified. Was it commercial or synthesized? If commercial please mention it, if synthesized, please cite the synthetic paper or add the NMR-spectra for the 81 compounds which prove the characterization of these compounds.

We believe that the answer to this concern has already been provided above. Please see first set of answers provided at the beginning of these paragraphs.

- It would be interesting to evaluate the activity of best compounds against resistant-strains which would make these compounds more attractive.

This was already responded above. Unfortunately we do not have these strains in our lab to carry on with the tests. We have started contacts to import them and be better prepared to arrange such experiments in the near future.

- The authors should perform a docking study to show what the target of these compounds and what is the binding affinity; could be N-hydrolases (NH), purine nucleoside phosphorylase (PNP) and also 5′-deoxy-5′-methylthioadenosine phosphorylase (MTAP).

New sections on the docking study are now part of the re-submitted manuscript. They are shown in Methods, Results and Discussion entries. To summarize obtained results of this in silico study, there are two new tables (see Table 3 and Table 4 in the resubmitted version) showing the docking results and a new figure (see Figure 7 in the resubmitted version) showing PyMol visualization of the most relevant targets. Additionally, a new Supplementary table (Table S3) with relevant information of P. falciparum and T. cruzi enzymes used for the docking study has also been incorporated.